# Position: Robust AI Personalization Will Require a Human Context Protocol

Anand V. Shah[1]   Tobin South[2]   Talfan Evans[3]   Hannah Rose Kirk[4]   Jiaxin Pei[2]   Andrew Trask[5]
E. Glen Weyl[6]   Michiel A. Bakker[1]

## Abstract

Personalization underpins the modern digital economy. Today, personalization is largely implemented through provider-managed infrastructure that infers user preferences from behavioral data, with limited portability or user control. However, large language models (LLMs) are increasingly being used to perform tasks on users' behalf. The age of LLMs for the first time provides a path to a more controllable and interpretable personalization paradigm, grounded in user-expressed natural language preferences and context. In this position paper, we argue that **to provide robust and user-centric personalization, we need a new Human Context Protocol (HCP) to represent and share personal preferences across AI systems**. HCP treats preferences as a portable, user-governed layer in the personalization stack, enabling interoperability, scoped access, and revocation. Along with a working prototype to ground discussion, we consider counterarguments along adoption dynamics and market incentives, high-stakes use cases, and outline novel paths via the HCP towards trustworthy personalization in the human-AI economy.

## 1. Introduction

Large language models (LLMs) are rapidly becoming embedded in everyday digital experiences, transforming how people access information and services. Central to unlocking their full potential is "personalized alignment" – tailoring model behavior to reflect individual preferences, values, and contexts (Kirk et al., 2024). This evolution toward personalization is accelerating rapidly, with major AI providers including OpenAI (2025), Google (2025), and Meta (2025) having announced personalization features in 2025 as central axes of their development roadmaps.

However, the current paradigm for personalization presents significant challenges. First, preference data is often opaque and incontestable – while personalization is predicated on knowing user preferences, users rarely see what the system "knows," exacerbating problems of privacy, and of shallow or inaccurate personalization (Kleinberg et al., 2024). Second, preference data is often non-portable, reinforcing user lock-in and harming market competition. Context cannot move easily across models or services, which raises switching costs and stymies downstream interoperability (Farrell & Klemperer, 2007). Both challenges reflect deeper questions about user ownership and portability of preference data, and about the thin, provider-dominated market for personalization infrastructure.

Recent initiatives like the Model Context Protocol (MCP) aim to create open standards for connecting AI assistants to wide-ranging data sources (Anthropic, 2024). While valuable for standardizing access to context, MCP does not address questions of ownership, granular user control, or privacy management for personal preferences. Yet these questions are vital to deployment.

We argue that **a dedicated, user-centric layer for preference management is a core requirement for building AI systems that are genuinely personal, interoperable, and aligned with diverse human values.** Market incentives alone will not produce this infrastructure, as current providers have weak incentives to enable portability or relinquish custody of preference data. Good deployment requires the ML research community to establish design principles, study safety risks, and inform emerging standards. To this end, we propose the Human Context Protocol (HCP), a system in which user preferences are managed by a dedicated intermediary – e.g., an LLM – that serves as the interface between individuals and the AI systems acting on their behalf.

Concretely, an HCP should enable individuals to:

- **Control access** to their preferences across LLM-powered services through fine-grained, revocable, and

[1]Massachusetts Institute of Technology, Cambridge, MA, USA [2]Stanford University, Stanford, CA, USA [3]Cursive, USA [4]University of Oxford, UK AI Security Institute, Oxford, UK [5]OpenMined, USA [6]Microsoft Research, USA. Correspondence to: Anand V. Shah <avshah@mit.edu>.

*Proceedings of the 43$^{rd}$ International Conference on Machine Learning*, Seoul, South Korea. PMLR 306, 2026. Copyright 2026 by the author(s).

purpose-scoped permissions;

- **Port preferences** across models and providers, reducing switching costs and mitigating lock-in; and

- **Actively shape** how preferences inform model behavior via clear, in-context elicitation and correction loops.

The paper proceeds as follows. In §2 we describe the background and related work. In §3, we propose the HCP. In §4 we consider alternative views and counterarguments. §5 presents a call to action and §6 concludes.

## 2. Background and Related Work

The conversation on digital personalization often begins with the countervailing right to privacy. This tension between privacy and personalization has driven successive waves of theory and product for personal-data control. Initial discussions on privacy centered on personal dignity and the right to self-disclosure (Westin, 1968). Yet, as online data proliferated in the computer age – often invisibly and at immense scale – this individual control was increasingly undermined, leading digital scholars to expand the frame of privacy to include protection from commercial exploitation (Laudon, 1996; Varian, 1996) and inspiring designers towards architectures that prioritize agency.

### 2.1. Work on Personal Data

The modern genealogy of user-controlled data begins with Hagel and Rayport's 'infomediaries,' imagined brokers that would negotiate data use on the individual's behalf (Hagel III & Rayport, 1997). Although visionary, infomediaries never overcame the two-sided-market adoption barrier, requiring buy-in from both users and firms in a time where internet markets were still nascent.

A more durable ideological basis for personal data control emerges in movements like Europe's MyData, which articulated human-centric principles such as portability and individual data sovereignty (Poikola et al., 2015). Tim Berners-Lee's Solid project operationalized similar ideals in "pods" – decentralized architectures where users store data and manage access via revocable permissions (Sambra et al., 2016). The Self-Sovereign Identity (SSI) movement extended this logic to digital identity, arguing that identifiers should be user-controlled rather than issued or maintained by central authorities (Allen, 2016; Mühle et al., 2018). More recent implementations (particularly Web3-enabled "data wallets") extend this model further, aiming to give users custodial control over identity, reputation, and other personal data using cryptographic methods (Zyskind et al., 2015). While there has been much work on building independent personal data stores, these efforts have yet to yield a widespread user-controlled preference management solution.

Recent advances in AI may change this history in two material ways. First, the value proposition for users contributing preference data has increased substantially. User data now supports increasingly capable AI systems that function as general-purpose assistants, and preference data further personalizes AI systems to the user themselves (Ouyang et al., 2022; Poddar et al., 2024; Sorensen et al., 2025).

Second, the emergence of natural language as the primary interface modality for AI systems substantially reduces the cost of expressing and updating preferences. Textual input offers a more accessible and natural means for users to articulate complex contextual information and preferences. A comparison of HCP to previous artifacts of personal data control are summarized in Table 1.

### 2.2. How Personalization is Done Today

Personalization in contemporary AI systems is implemented through mechanisms embedded within model-provider infrastructure. These mechanisms vary – post-training via RLHF or DPO encodes population-level behavioral priors into model parameters (Ouyang et al., 2022; Rafailov et al., 2023), often guided by explicit provider values like Anthropic's "helpful, honest, harmless" (HHH) framework or Constitutional AI (Askell et al., 2021; Bai et al., 2022); in-context specification and memory systems persist fragments of prior conversations; behavioral inference constructs implicit representations from interaction patterns using latent-variable models (Poddar et al., 2024; Li et al., 2024; Jaques et al., 2019; 2020); and explicit elicitation surfaces preferences through surveys or reflective dialogue (Blair et al., 2025; Handa et al., 2025). But across all these approaches, the same structural pattern recurs: preferences are formed and acted upon within provider systems, yet do not exist as durable, user-governed representations. Preference data remains internal to providers and does not carry across services.

This has economic consequences. Although users retain nominal rights to delete data or opt out of certain uses, providers typically maintain broad, perpetual licenses to use and sublicense user-provided information. In practice, this grants providers substantial de facto control over preference data – what Grossman & Hart (1986) would call residual rights. Users supply the inputs but do not control how those inputs are combined, interpreted, or deployed across contexts. This is an economic problem for the ML community to contend with: this position paper argues for preference infrastructure that would improve competition and user agency, and calls on researchers to guide the design discussion.

Nothing in this diagnosis requires a single representational format. What matters is that preferences exist as a distinct layer in the personalization stack – one that is user-mediated,

*Table 1.* Evolution of User Data Control.

| Initiative | Key Idea | Mechanism | Limitations |
|---|---|---|---|
| Infomediaries (Late 1990s) | Brokered user data via intermediaries | Third-party agents managing consent | Indirect control; user frictions; requires large market adoption |
| MyData (2010s) | Data sovereignty as a civic right | Normative principles | Lacked a specific technical implementation |
| Solid Project (Mid 2010s) | User-controlled decentralized storage | Data "pods" with revocable permissions | User frictions (self-hosting); ecosystem still developing; limited natural language scoping |
| SSI (Mid 2010s) | Portable, user-owned digital identity | DIDs and verifiable credentials | Limited to identity attestations; architecturally unsuited for rich data |
| Web3 Data Wallets (Late 2010s) | Custodial control over digital assets | Keys, smart contracts, blockchain | High user frictions; limited legal recourse (relies on "code is law"); asset-centric design |
| **HCP (current)** | **User-directed preference management** | **LLM-native preference interface** | **Adoption requires ecosystem buy-in; ensuring security & mediating LLM integrity is crucial** |

portable, and subject to explicit authorization. Natural language is the most obvious candidate: legible to users, interoperable, and native to language models. The next section discusses desiderata for such a system.

## 3. Human Context Protocol (HCP)

The goal of the HCP is to be an easy and reliable control layer that governs how AI systems access personal user context. Its defining feature is a dedicated intermediary which interprets user preferences and enforces scoped, minimal disclosure to downstream models at inference time.

### 3.1. Key Attributes

To realize the vision of an HCP, any system design should have the following core attributes:

- **Interoperability**: HCP must be interoperable across AI models and application contexts, as this is fundamental to its utility. Interoperability should be facilitated by open, well-supported, and existing communication protocols.

- **Encapsulation**: For HCP to provide genuine utility, it must be capable of richly capturing user preferences. While current solutions have limitations in preference elicitation and representation, the HCP's data model should leverage advances in preference representation

(whether as text, graph-based knowledge structures, vector embeddings), provided those representations are portable.

- **Control**: Given the personal and sensitive nature of preference data, users must have fine-grained, revocable, and editable control over what preference information is shared and with whom. For instance, a user should be able to share culinary preferences with a recipe generator without exposing mental health information. This aligns with the principle of data minimization (as in GDPR (2016) Article 5(1)c), ensuring only necessary information is disclosed for a given query.

- **Security**: The storage and transmission of sensitive personal data within HCP demands robust security measures. Preferences must be secured at rest and in transit, with strong authentication and authorization mechanisms to ensure AI models only access explicitly authorized preference subsets (South et al., 2025a).

### 3.2. System Design

This paper does not prescribe a definitive implementation for HCP; any system that satisfies the aforementioned design attributes would be suitable. However, to facilitate discussion, we outline a *potential* protocol architecture below. The subsections describe preference representation,

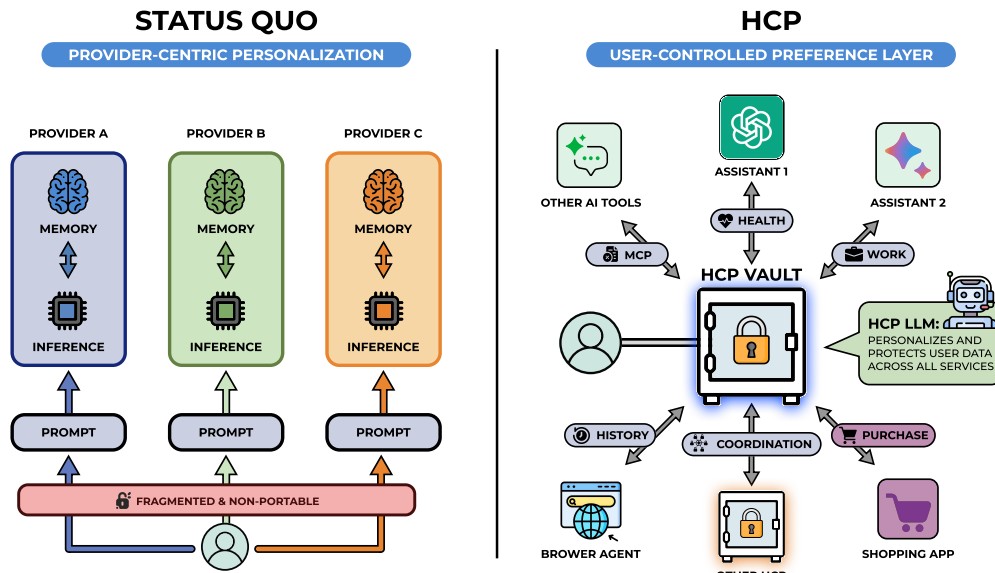

*Figure 1.* Illustration of two personalization paradigms. On the left panel, personalization is implemented within individual AI services, with user prompts and interaction data feeding service-specific memory and inference pipelines. As a result, preferences are non-portable across providers. On the right panel, a shared preference layer is introduced in which user context is stored in a centralized HCP vault and selectively accessed by multiple assistants, tools, and applications through an HCP-mediated interface. User preference data (generated by varied user activity) is moderated by HCP to consumer agents. Each agent obtains only the relevant subset of the user's complete preference data.

supported actions, initialization, and mechanisms for access control, security, and authentication.

### 3.2.1. CONTEXT REPRESENTATION AND STORAGE

One key aspect of the HCP system design concerns the representation and storage of personal context and preferences. Language models operate natively in text with high fidelity in both reading and writing. This makes text a well-suited paradigm for storing and editing preferences – unlike, say, large-scale recommender systems, which typically encode user preferences as task-specific embeddings learned implicitly from behavior and tightly coupled to a particular model or objective. This format aligns with the dominant input modalities of LLMs, offering native interoperability without requiring specialized serialization formats or custom embeddings. For example, preferences such as "I prefer Mediterranean cuisine" or "I enjoy movies directed by Christopher Nolan" are both human-readable and machine-usable.

To reduce ambiguity, preferences may be enriched with lightweight schema annotations (e.g., a category tag like `food` or a confidence score). Versioning should be supported to allow preferences to evolve over time while preserving historical context. Clarification protocols are also needed to let users refine vague or underspecified statements interactively (Pyatkin et al., 2022).

The underlying storage mechanism could vary: a single comprehensive document (e.g., JSON or Markdown) serving as a unified profile; a key-value store with attached access controls; a graph-based representation capturing relationships between preferences (Pan et al., 2024); or integration with personal data pods like Solid (Sambra et al., 2016). For larger datasets, vector databases supporting semantic retrieval can make scalability cheap.

### 3.2.2. ACTIONS

The HCP must support a complete lifecycle of preference operations. At a minimum, four types of actions are envisioned:

- **Read**: retrieving preferences on demand, possibly filtered by category or queried semantically (e.g., "What are the user's vacation preferences?").

- **Write**: adding new preferences, with policies for conflict resolution when overlapping entries exist.

- **Update**: modifying preferences while maintaining version history.

- **Delete**: removing preferences, with compliance guarantees for data erasure such as GDPR's "right to be forgotten."

These actions may be exposed as RESTful endpoints (`GET`, `POST`, `PUT`, `DELETE`) or as MCP tools callable directly by LLM agents. A central design question is granularity. One pragmatic design is to use general-purpose methods like `searchPreferences` and `updatePreferences`, with explicit scoping parameters (e.g., `category=food`) that control retrieval and authorization.

### 3.2.3. INITIALIZATION

Initialization determines how an HCP instance is created and begins managing personal context. Bootstrapping could draw on onboarding interviews, user-supplied documents, software integrations, or imported digital traces such as playlists and browsing history. Over time, the preference corpus should evolve through continuous updates, balancing recency with user control. This is similar to how many existing application-specific memory systems work for chatbots.

The HCP LLM plays a central role during initialization and beyond. This model, potentially smaller, locally hosted, or specialized, begins by making common-sense decisions to be later refined or with a minimal user intake process.

### 3.2.4. ACCESS CONTROL, SECURITY, AND AUTHENTICATION

Given the sensitivity of user preference data, an implementation must enforce strong guarantees of access control, confidentiality, and integrity.

**Access Control** Users hold fine-grained, revocable permissions. A recipe generator may receive dietary preferences but not unrelated (and potentially sensitive) medical data. Whether an agent receives a category can be settled by a deterministic permission check (e.g., an OAuth layer over an MCP) (South et al., 2025b). Permissions may be granted for a single request or as a standing arrangement, and can be revoked at any time; this revocation propagates downstream, at times forcing deletion (Trask et al., 2020).

**Security** Preferences are encrypted in transit and at rest, and an audit log records which downstream agents read preference data and when. Further security preferences are the user's to set. For high-stakes categories like payment credentials or medical data, entries can be further encrypted under a password only the user holds, requiring a human unlock at the moment of use. Routine categories can default to direct LLM mediation. The tradeoff between assurance and friction is therefore the user's to make, not the provider's, and any category can be moved to the stricter, ask-first tier.

**Authentication and Integrity** Digital signatures protect stored preferences against tampering, and the permission check described in *Access Control* ties every access to a prior user authorization. Importantly, these guarantees are conventional and verifiable with existing tools, as is the cryptographic unlock path for high-stakes categories. In addition, HCP must anticipate emerging threats in LLM contexts. For example, inference attacks that reconstruct hidden preferences from observable outputs; adversarial prompting, where malicious models attempt to elicit oversharing; and subtle manipulation of smaller HCP LLMs by more capable external systems. Safety from these emerging threats is itself a capabilities frontier along which HCP implementations will compete.

### 3.2.5. DEMONSTRATING FEASIBILITY: AN OPEN-SOURCE PROTOTYPE

The proposed conceptual design is readily implementable, a crucial characteristic for fostering an open preference ecosystem. To demonstrate viability and encourage further work, we developed an open-source proof-of-concept.[1]

This prototype embodies several core attributes. It is a web application where users manage preferences and control third-party access. An integrated MCP server supports interoperability with compatible AI interfaces, enforcing user authentication and granular authorization for distinct preference categories. For simplicity, this demonstration omits the orchestrating LLM – instead relying upon access requests passed through MCP to determine what information is shared with the user. More complex implementations, which ingest existing personal context and address the cold start problem to route relevant information, are also available.

While an early step, this prototype confirms the feasibility of constructing an HCP that is interoperable, secure, and grants users meaningful data control. It provides a foundational codebase for the community to build upon.

### 3.3. Example

**Concrete user scenario.** *Alex* uses a Claude-based assistant on their phone. They install the HCP integration by adding the HCP–MCP server from the assistant's integrations marketplace and completing an OAuth 2.0 consent flow that grants *category-scoped* access to `outdoor_gear` and `health_context` only. During a chat about hiking, Alex mentions: "I need to make sure my boots support high arches." The assistant recognizes this as a standing constraint and invokes `addPreferences` on the HCP MCP tool, which stores a natural-language entry under `health_context` with provenance (time, source message, model).

---

[1]Demo repository: https://github.com/avshah1/hcp-demo

Two weeks later, Alex asks: "What lightweight boots should I buy for the Pacific Crest Trail?" The assistant calls `searchPreferences`. The *HCP LLM* evaluates the request against the authorized categories, retrieves only the minimal relevant snippets (e.g., the "high arch support" constraint), and returns a scoped preference bundle. The assistant may then query external product catalogs filtering for arch support and compose a response. Throughout, HCP enforces least-privilege, maintains an audit trail, and allows Alex to revoke scopes or edit entries at any time.

**Failure case.** Suppose the assistant also asks about Alex's budget. If Alex never granted access to `financial_context`, the HCP returns nothing for that category – the assistant cannot see budget constraints Alex chose not to share. Throughout, HCP maintains an audit trail, allowing Alex to review what was accessed, grant or revoke scopes, or edit entries at any time.

**End-to-end flow.**

1. **Setup.** User enables HCP via MCP integration (Anthropic, 2024); OAuth 2.0 consent is issued with category-scoped grants (Hardt, 2012).

2. **Capture.** Assistant tool-call `addPreferences` sends {category=`outdoor_gear`, NL text, metadata} to HCP.

3. **Persist.** HCP validates token and scope, normalizes/versions the NL entry, and stores it in the preference store (with optional vector index).

4. **Query.** Later, assistant invokes `searchPreferences` for "shoes for the Pacific Crest Trail."

5. **Minimize.** HCP validates scope; the HCP LLM selects only relevant items from authorized categories and redacts unrelated fields.

6. **Compose.** Assistant may call external product APIs or A2A frameworks (Google, 2025) using only derived, minimized preference facts.

7. **Explain & Log.** Response includes rationale and (optionally) HCP-provided citations; HCP appends an auditable record. User can view, edit, or revoke.

**Interoperability note.** While the example uses MCP tooling for assistant integration and OAuth 2.0 for authorization, the same flow applies with alternative agent-to-agent transports (Google, 2025) or assistant runtimes. The key invariant is that HCP mediates preference access, applies data minimization at inference time via the HCP LLM, and preserves user control through explicit, revocable scopes and auditable operations.

## 4. Alternative Views

This paper argues that designers ought to build a preference layer that exists on top of (and independent of) large model providers. In this section, we consider credible counterarguments to this position.

### 4.1. The Coasian Objection: Does Architecture Matter?

A natural objection is that it doesn't matter who controls preference data – markets will sort out its use efficiently. Coase's theorem states that under zero transaction costs and well-defined property rights, markets achieve efficient outcomes regardless of initial allocations (Coase, 1960). If this held, HCP would be a convenience feature rather than welfare-relevant infrastructure.

However, several market failures mean that who controls preference data does matter for outcomes.

#### 4.1.1. WHY THE COASIAN BENCHMARK FAILS

Below, we describe some particular market failures which refute the intuition from Coase's theorem.

First, **lock-in and interoperability** constitutes a failure of the zero transaction cost assumption. When preference data is locked within specific service silos, users face high switching costs if they wish to employ a competing agent or service. This friction limits user choice and dampens competitive pressure on agent providers to improve quality or compete on price (Farrell & Klemperer, 2007). For example, prior to phone number portability regulations in telecommunications, switching carriers also meant losing one's number – a critical piece of digital identity. The introduction of number portability dramatically increased competition and reduced prices (Viard, 2007). A second illustrative example comes from financial services – prior to open banking regulations, consumers' transaction histories were locked within incumbent banks, creating high switching costs and limiting entry by new financial intermediaries. Open banking regimes, such as Europe's PSD2, mandated standardized, user-authorized access to account data, enabling third-party providers to compete on services while banks retained custody of funds. Empirically, these reforms increased fintech entry and competition (Babina et al., 2025). Similarly, HCP, designed with interoperability as a core principle, would reduce switching costs and foster a more dynamic ecosystem in which agents compete on performance.

Second, **the non-rival nature of (preference) data** constitutes a failure of property rights. Unlike physical goods, data is non-rival – its use by one entity does not diminish its availability for others. This characteristic implies that social welfare is maximized when valuable data is used broadly, subject to privacy constraints. However, when firms con-

trol user data, competitive incentives lead to inefficient data hoarding; firms are reluctant to share data that might empower rivals or accelerate their own creative destruction (Jones & Tonetti, 2020). HCP, by assigning control to the user, provides a mechanism to ameliorate this market failure – users can choose to license their preference data as broadly as is useful for themselves, enabling the aggregate productivity gains typically associated with information goods.

Third, **information asymmetries and market power** constitutes a departure from perfect competition on the firm side. Large firms often possess far more information about market conditions and user behavior than individual users do, along with the analytic tools to exploit that asymmetry. For example, (Acquisti & Varian, 2005) describe precisely this dynamic in data markets – firms extract user surplus by leveraging purchase history to conduct targeted pricing. By giving users control over the release of their preference history and associated information, HCP empowers consumers to strategically manage (exploitation from) their information footprint.

Fourth is a concern of servicing **diverse preferences among users**. This is a failure of market thickness. When users have heterogeneous preferences – particularly regarding privacy, ethics, or cultural norms – market-based solutions tend to systematically underserve those with non-mainstream preferences (Waldfogel, 2003). HCP addresses this failure by empowering all individuals to define and enforce their own specific preference boundaries through granular controls, ensuring their values are respected regardless of the prevalence of bespoke market solutions.

### 4.2. The Revealed Preference Objection: Why Not Just Infer?

A second objection holds that behavioral inference is sufficient – explicit preference articulation is unnecessary because preferences are "revealed" through choices (Samuelson, 1938). This justifies inference over actions, an enormously powerful paradigm given bountiful user action data.

However, modern behavioral economics (and common sense) yields many examples where this paradigm fails – problems of mental accounting (Thaler, 1985), self-control (Thaler & Shefrin, 1981; Laibson, 1997), and the difficulty of inference for complex objectives. Current personalization systems rely almost exclusively on behavioral proxies – clicks, time spent, purchase history – with few in-roads for direct expressions of user intent.

Exclusive reliance on inference creates misalignment between what systems optimize for (proxies) and what users actually want. A news recommendation system might interpret clicking on sensationalist headlines as a preference for such content. (Kleinberg et al., 2024) call this the "inversion

problem": systems must work backwards from observable actions to infer mental states. HCP addresses this by providing a mechanism for **direct preference articulation** to supplement and ground inference – particularly valuable for complex preferences difficult to infer from behavior alone, such as privacy boundaries, ethical values, or content standards. Moreover, HCP can support opt-in sharing of preference categories across users who consent, improving the data available for inference while preserving user control.

### 4.3. The Model-Level Objection: Why Not Fine-Tuning or Steering?

A third objection is that personalization can be achieved at the model level – through fine-tuning, adapters, or steering vectors – making dedicated preference infrastructure unnecessary. If we can align models to human preferences through training, why do we need a separate protocol layer?

The limitation is that these are provider-side solutions. A user's personalized adapter for Claude cannot transfer to ChatGPT. These approaches solve personalization *within* a provider's ecosystem but not *across* providers. They also keep preference representations internal to provider optimization pipelines – users cannot inspect, edit, or revoke the learned preferences.

HCP addresses this by making preference data portable and user-mediated. Moreover, it complements model-level personalization: users express preferences once, and those preferences can be used to further personalize any base model. This also enables what Sorensen et al. (2024) call 'steerable pluralism' – models reflecting user-defined viewpoints at inference time, without additional per-user fine-tuning.

### 4.4. Practical Challenges and Limitations

We acknowledge several practical challenges that must be addressed for successful implementation.

#### 4.4.1. STANDARDS CONVERGENCE

The core obstacle is standards convergence. Multiple vendors must agree on a stable interface for declaring, storing, and exchanging preferences, yet the pace of model innovation makes any rigid specification brittle. Successful precedents – from TCP/IP to HTML to OAuth – show that interoperability wins when standards are open, modular, and versioned, letting new capabilities slot in without breaking legacy clients (Clark, 1988; Simcoe, 2012; Hardt, 2012; Ghazawneh & Henfridsson, 2013). We believe academic discussion here would be particularly useful in guiding industry towards appropriate standards.

A related concern is bootstrapping adoption. Even if designers determine an optimal standard, incumbents treating

preference data as a competitive moat are unlikely to adopt HCP without compelling incentives.

Yet the market may provide its own solution. Password managers offer a useful precedent: users store credentials in a third-party layer rather than with each service; services did not need to explicitly opt in, as password managers work via browser integration; and adoption began with power users and expanded as friction decreased.

Unlike password managers, HCP requires some integration (MCP or API access) rather than passive autofill. But the wedge may be similar: start where integration already exists (MCP-compatible assistants), prove value in specific domains, and expand. For example, a scheduling HCP could accumulate user context over time (availability rules, relationships), then extend to adjacent tasks (trip planning, reminders), eventually evolving into a general-purpose preference management tool. Complementarily, academic designers drive standards convergence across products and anticipate deployment risks as the market matures. This approach constitutes one realistic rollout story by which competition solves the standards problem.

#### 4.4.2. Risks from Deep Personalization

The very capability that makes personalization valuable – enabling AI systems to adapt to individual preferences – also gives these systems increased purchase on users' lives and decisions. This may magnify risks from bad actors, who could use personalization for manipulation or belief persuasion.

Beyond malicious use, user inconsistency also creates direct concerns that require careful oversight. First are off-target effects from **information asymmetry**: users may overlook how a system actually affects their psychology, with recent evidence from sycophancy (Sharma et al., 2025; Fanous et al., 2025). Second, are concerns from **present bias**. Users may use AI products myopically, becoming dependent on them to the detriment of their future well-being.

#### 4.4.3. Ethics and Oversight

Finally, there are also some ethical considerations to note in the (long-term) deployment of HCP.

- **Digital-divide mitigation.** If HCP is usable only by sophisticated or affluent users, it risks widening existing inequities in realizing the benefits of technology.

- **Accountability frameworks.** A user-centric architecture needs transparency requirements, audit mechanisms, and accessible dispute-resolution processes to address violations.

- **Social nature of data.** Preferences often have

shared or networked ownership; HCP may want to include governance mechanisms that respect overlapping claims on preference subsets.

We view each of the difficulties listed in this section not as insurmountable obstacles, but as research questions worthy of collaborative effort.

## 5. Call to Action

We outline directions for advancing user-controlled preference infrastructure. Appendix A provides extended discussion of applications across domains including scheduling, health, education, and democratic governance.

**For ML researchers.** This paper outlines design directions for user-mediated preference architecture, and the discussion of this design is itself a call to action given academic discussion often informs and inspires downstream product design. However, beyond dissemination, there are several concrete research directions. First, safety: deep personalization creates risks – manipulation, sycophancy, dependency – that need study before widespread deployment. Second, engineering problems: how to elicit complex preferences, how to represent them portably, and how to secure them against adversarial queries. Third, evaluation: how should benchmarks for personalization quality be designed, and how might newfound user preference data help? Such benchmarks may also be useful for safety in measuring the purchase of persuasive or manipulative systems.

**For policymakers.** GDPR mandates data portability, but for AI preferences this right remains largely unexercised – HCP offers infrastructure to make it exercisable. Beyond compliance with existing policy, policymakers will need to define what portability means for AI systems: whether services must accept preference imports, in what formats, and with what guarantees. Defining these standards also creates safe harbors for builders who comply. Finally, preference management may be deployed in a variety of contexts. For example, HCPs may serve as custodial mechanisms for content – parents managing preferences over what their children interact with – where regulatory frameworks are particularly high-impact (see Appendix A).

## 6. Conclusion

As generative AI technologies become more capable and widespread, the mechanisms for personalization become increasingly consequential. We have argued that current approaches – where preferences are inferred rather than expressed, controlled by providers rather than users, and fragmented across services rather than portable – fail to realize the full potential of personalization while introducing significant risks of manipulation, user disempowerment, and

lock-in.

**Who controls preference data matters.** HCP offers a path forward: portable preferences, user-governed access, and interoperability without lock-in. Realizing this requires effort across research, policy, and adoption – but the potential for a pluralistic, user-empowered AI ecosystem justifies that coordination.

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

# A. Use Cases: Extended Discussion

## A.1. Expanding User Agency: From Preference Expression to Discovery

Many industries across the information economy rely on preference formation and learning. While many models assume stable, well-defined preferences, evidence shows that consumers engage in costly search and experimentation to learn what they actually want (DellaVigna, 2009). Work on experience goods highlights that preferences for many products cannot be evaluated without trial (Nelson, 1970), and research on constructive choice shows that preferences are often formed during decision-making rather than simply retrieved (Bettman et al., 1998).

HCP enables **systematic preference discovery** through controlled self-experimentation across AI systems and contexts. Instead of treating preferences as static settings, users can try alternative preference profiles, observe downstream behavior, and iteratively refine what they endorse. This has precedent in the Quantified Self movement (Swan, 2012), but HCP extends the idea from physiological tracking to values, information diets, and decision rules.

**News and information.** Users can experiment with preferences over breadth vs. depth, source diversity, topic mix, and framing. This is especially useful when users are unsure what informational style helps them (e.g., short summaries vs. long-form analysis). Users can also test different lenses for the same topic (including political lenses) to better understand their own reactions and priorities (Druckman & Lupia, 2000). Critically, this does not have to be tied to any single domain: HCP can store multiple *time-indexed*

profiles ("what I wanted at 20, what I want now") and let users compare outcomes against their own past baselines, making preference change legible rather than implicit.

**Matching markets.** Many matching markets clear based on preferences under search frictions, and better preference clarity can improve outcomes. Two salient examples are dating (Rosenfeld & Thomas, 2012; Finkel et al., 2012; Hitsch et al., 2010) and school placement (e.g., college admissions or the NRMP match) (Roth & Peranson, 1999; Gale & Shapley, 1962). HCP can help users articulate and update constraints (dealbreakers, tradeoffs, long-run vs. short-run priorities), then test how those choices change suggested matches or application strategies, without rebuilding the preference profile from scratch each time.

**Product discovery.** Most obviously, the global digital advertising industry – worth over $600 billion annually (eMarketer/Insider Intelligence, 2024) – is built around reducing search costs and helping users discover products. HCP allows a more user-directed version of discovery: users can specify what kinds of products and pitches they want to see, what they want to avoid, and what tradeoffs they want to optimize (price, sustainability, quality, novelty). The same mechanism can cover entertainment discovery (e.g., what to watch or listen to) as a lower-stakes sandbox for testing and refining preferences.

In general, HCP's preference-discovery value is highest for *high-dimensional* preferences, in markets with substantial product diversity (where expansive search is costly), and in one-shot, high-stakes scenarios – precisely the settings where today's personalization is hardest to control.

## A.2. Novel Downstream Mechanisms: Building on the Preference Layer

Standardizing preference expression and management creates a foundation upon which entirely new mechanisms can develop, much as standardized protocols enabled the flourishing of internet applications by reducing transaction costs and enabling new goods and services (Shapiro & Varian, 1999).

The key insight is that when preferences become structured, portable, and machine-readable, they can serve as inputs to coordination mechanisms that were previously impractical due to high transaction costs. Moreover, the mechanisms themselves can include new types of commitment. This enables everything from sophisticated group decision-making to collective bargaining structures that aggregate individual preferences into coordinated action. In practice, these mechanisms are typically executed by user-side assistants ("personal agents") acting on the user's behalf across services, and will only continue to do so as transaction costs continue to decrease (Shahidi et al., 2025). Below, we out-

line several illustrative applications that demonstrate HCP's potential to enable novel forms of digital cooperation and governance.

**Guardian assistant systems.** Perhaps the most immediately valuable application is the creation of "guardian assistant" layers – middleware AI systems that sit between user HCPs and other digital services. These guardians, operating with full access to user HCPs, serve a crucial dual function. Primarily, they act as digital advocates to enforce a user's own preferences. However, they also serve as a control layer, implementing policies set by trusted third parties that can supersede a user's immediate intentions, either for the user's own protection or to prevent harm to others.

Such guardians could intercept outbound prompts and inbound content to identify persuasive tactics or deceptive patterns, flag potential manipulation attempts based on known user vulnerabilities, filter content, add browser overlays that provide relevant context, and negotiate automatically with third-party systems based on user-defined boundaries. This protective function is especially vital for children, where a parent's policies for content filtering can override a child's immediate choices to shield them from harmful material.

Furthermore, while much of this paper situates the HCP as a user-specific technology, it's important to note that the 'guardian' can also be used to reflect more complicated social relations. In particular, consider the relation between an employer and employee. A user's calendar data may reveal private information about their employer. To manage this risk, the user's firm may wish to be guardian to their network of employees, overseeing user-specific data scoping to ensure that sensitive firm-specific information isn't accidentally leaked.

**Group coordination mechanisms.** When individual preferences are structured and accessible, new possibilities emerge for group decision-making that go far beyond simple polling or majority voting. Tools built atop HCP could aggregate compatible preferences to facilitate coordination problems ranging from scheduling to collaborative project planning. Unlike traditional voting systems, these mechanisms could perform sophisticated preference matching, identifying complementary patterns and potential compromises that satisfy multiple constraints simultaneously (Tessler et al., 2024; Bakker et al., 2022).

Consider planning a group activity where participants have expressed different primary preferences. The system might recognize that while Alice prefers outdoor activities and Bob prefers cultural events, both share a secondary preference for novel experiences – suggesting an outdoor cultural festival as an optimal compromise. This capability explicitly plays out the analogy of revelation mechanisms from economic theory, but with dramatically reduced transaction costs due

to the structured preference data that HCP provides. Additionally, it can also be paired with other discursive methods, to enable clearer debate and value negotiation (Burton et al., 2024).

**Negotiation, collective action.** HCP enables users to pool preferences into cooperative structures that can exercise collective leverage, directly addressing fundamental power imbalances in digital markets where individual users face large platforms. Consider a preference cooperative focused on privacy practices: members contribute their privacy preferences to a shared layer, with an agent that negotiates with services on behalf of the entire group. Services might offer improved terms to access this aggregated market, similar to how buying clubs achieve volume discounts through coordinated purchasing power.

This collaborative approach creates collective mechanisms for users to resist surveillance practices and reclaim agency in digital environments. Such preference pooling could extend across domains: negotiating improved service terms or features, coordinating responses to services that violate common preference boundaries, facilitating data unions that derive shared value from combined preference data, and creating preference-based mutual aid networks where compatible preferences enable resource sharing.

**Public services.** Public-sector services are a natural fit for HCP-style preference infrastructure because they often require personalization while operating under strong constraints around consent, purpose limitation, and accountability. Across domains like education, healthcare, and benefits administration, individuals repeatedly re-specify the same constraints (needs, communication preferences, accessibility requirements, eligibility-relevant context), and each institution ends up maintaining its own partial, non-portable profile. HCP offers a user-governed way to carry these preferences across services, while still enabling fine-grained, purpose-scoped sharing and revocation.

Educational institutions could provide HCP infrastructure as a complement to the digital investments (e.g., laptops or tablets) made in schools. This would let students (and, where appropriate, guardians or school administrators) maintain a persistent set of learning-relevant preferences – such as accessibility accommodations, language support, pacing, and feedback style – and share only the relevant subset with specific tools or vendors. Different pedagogical approaches could then be implemented through how institutions and educators choose to request and apply these preferences, while students retain a clear, inspectable record of what is being shared and why. This enables personalized learning without forcing each platform to build its own closed preference silo.

**Democratic governance.** HCP could support democratic

governance by enabling citizens to share *purpose-scoped* preference profiles with elected representatives or public institutions. The point is not to replace elections (or deliberative mini-publics), but to complement low-frequency, low-bandwidth voting with higher-resolution, consented signals about priorities, tradeoffs, and constraints on specific policy areas. In this sense, HCP would let governments form more fine-grained representations of public will than crude polling or whoever is loudest in public comment channels, while preserving privacy through explicit user authorization.

Concretely, HCP could enable new aggregation and interaction patterns. For example, governments could run issue-specific consultations where citizens contribute (or selectively reuse) relevant preference snippets, then apply clustering and sensemaking methods to identify stable coalitions and common-ground statements at scale (as in Polis-style opinion mapping) (Small et al., 2021; Konya et al., 2025). Systems that use HCP can also feed deliberative processes by helping select agendas, generate structured alternatives, and surface value conflicts for deeper discussion, rather than treating deliberation as a one-off event (Fishkin & Luskin, 2005).

### A.3. Broader Societal Implications

The widespread adoption of HCP (and novel, downstream mechanisms) would likely trigger significant second-order effects across markets, policy, and governance. These implications extend far beyond individual personalization to reshape how digital ecosystems organize around user agency.

**Market evolution and new economic structures.** Just as app stores emerged atop standardized mobile operating systems, HCP would likely spawn markets for specialized preference management tools, guardian systems, and preference-discovery services. A natural evolution would be the emergence of a "marketplace of licensed guardians" – specialized AI systems certified to protect user interests in specific domains.

These might include **child safety guardians** that enforce age-appropriate interactions based on parent-defined HCP layers, **financial guardians** that protect against manipulation in high-stakes transactions, **health guardians** that ensure medical AI systems respect patient treatment preferences and risk tolerance, and **professional guardians** that maintain workflow preferences while protecting against distractions.

The key idea is that many user contexts have society-approved mores associated with them, but that users' preferred solutions may differ. Such marketplaces would create powerful incentives for innovation in preference protection and enhancement.

**Policy development and regulatory frameworks.** HCP

represents a practical demonstration of data portability and user control that could inform future policy development across multiple domains. By showing that meaningful user control is technically feasible, HCP provides policymakers with a concrete reference model for regulations concerning data rights and AI governance. Current regulatory frameworks like GDPR include data portability requirements, but these remain largely theoretical without practical implementations. HCP offers a template for "by-design" approaches to regulation (Mulligan et al., 2016) – embedding policy objectives directly into technical architecture rather than imposing them through external compliance requirements.

**Distributed governance models.** The preference layer architecture suggests a novel model for distributed governance of AI systems, where control is exercised not through centralized oversight but through the aggregated preferences of users themselves. This approach aligns with concepts of regulation by architecture, where technical design choices enforce normative objectives (Lessig, 2009). Rather than relying solely on top-down regulatory intervention, HCP enables bottom-up governance through collective user agency – a form of technological democracy where the architecture itself becomes a mechanism for expressing and enforcing societal values about AI behavior.

