# OpenReview forum: "Position: Robust AI Personalization Will Require a Human Context Protocol"
_ICML.cc/2026/Position_Paper_Track — ICML 2026 Position Paper Track regular_

### Official Review · Reviewer_9ZGe · 2026-03-10

**Significance:** 4
**Argument Clarity:** 3
**Rating:** 6
**Confidence:** 4

**Questions:**

1. I realize this is not specifically a question about the position argument itself, but I wonder how preferences are captured? How conflicts (e.g., context-dependency amongst preferences) are dealt with.
2. The authors state that a model-based HCP could make common-sense decisions to minimize user intake. Would this be wise? I wonder about models stereo-typing users, and edge-case uses may not be represented by “common-sense”.

**Alternative Views Section:**

Yes

**Compliance With Llm Reviewing Policy A Conservative:**

Affirmed.

**Discussion Potential:**

4

**Paper Summary:**

The paper proposes changes to the way that machine learned models are personalized. In particular, the authors highlight the main issues with the current forms of personalization, especially closed frontier models. In particular, these methods are opaque, the user has no control over what is assumed about them, and the preferences are not transferable across models. This makes it difficult for users to control what is personalized about them and how this information is used.

To remedy this the authors propose the notion of a *human context protocol* that operates between the user and the underlying model such that the user has control over what the model is able to use in any given context, and the protocol offers a portable representation of the user that works across models (akin to model context protocols for transferable skills).

**Position:**

Yes

**Position In Title:**

Yes

**Related Work:**

4

**Strengths And Weaknesses:**

**Strengths**
- The concept proposed in this paper is novel and an interesting take on personalization. I realize novelty need not be a factor here, but I think that this is clear strength of the paper.
- If the notion of an HCP were to garner interest within the community, this would be a compelling way of providing control to user.
- The position of the authors is clear and put across well. I found this to be compelling.
- The extended discussion in the Appendix highlights the benefits and importance of the HCP concept — it has broad application and clear benefits.
- The paper considers the interdisciplinary nature of the problem, including from the perspective of users, developers, practitioners, and lawmakers.

**Weaknesses**
- The main weakness I see with the position adopted here is the resistance of the major players to relinquish control over how they collect and exploit user data.
- I realize that this is not specifically the “position”, but the proposal is a model that gates access to preferences given a request from a model for user-data. Thus the user is delegating responsibility for access to their preferences to the “HCP” model. I wonder why not just present the request for information directly to the user so they can approve/deny given the context. The reason I worry about this is that preferences are highly context-dependent, and often are either underspecified, or difficult to articulate.

**Support:**

4

---

> ### Author Rebuttal · Authors · 2026-03-31
>
> Thank you for the review. The points you raise about context-dependency and stereotyping risk are exactly the kind of design questions we hope the community will engage with. We respond to your questions below.
>
> **Q1 + W3: How are preferences captured? Aren't they too context-dependent and hard to articulate?**
>
> Preference data could enter HCP via onboarding interviews, user-supplied documents, decision traces under other software or MCP integrations, or imported digital traces such as memories from model providers or browsing history from search engines (§3.2.3). The main practical route is likely the last — providers already infer rich preference states from user interaction data, and HCP gives users a way to import, inspect, and make those inferences portable across services rather than leaving them siloed.
>
> Further, while we agree preferences are sometimes hard to articulate, one reason why we think the friction to entry on personal data stores has gone down is because of how easy natural language is for both users and models. E.g., "I generally prefer window seats but not on red-eyes" is hard to express as a structured field but fairly trivial as text. And, as model capabilities continue to improve, we expect that the cost of expressing, parsing, and managing this preference data continues to fall.
>
> **Q2: Should a model-based HCP make "common-sense" decisions? Risk of stereotyping?**
>
> Our response has two parts.
>
> First, we agree that "common-sense" inference decisions themselves carry a risk of bias. However, this risk is not unique to HCP — every personalization system makes such inferences. The difference is that in current systems, these inferences are often opaque and non-editable by users directly. In contrast, the HCP surfaces preferences as legible natural-language statements that users can directly inspect, edit, or delete (§3.2.4), likely improving visibility on incorrect stereotypes.
>
> Second, the "common-sense" initialization described in §3.2.3 is explicitly provisional. The HCP LLM makes coarse initial user inferences as a starting point to avoid the cold-start problem but has many avenues for self-correction.
>
> **W1: Major players will resist relinquishing control over user data**
>
> This may well be true. However, we think this resistance is less of a blocker than it might seem. We have three arguments here.
>
> First, HCP does not require downstream agents like scheduling assistants or shopping agents to cooperate. Such agents querying a user's HCP do not care whether preferences came from a provider's memory or from HCP — they simply need the preferences. Adoption can therefore begin where integration already exists (MCP-compatible agents) without model provider buy-in. A more difficult dependency is indeed upstream with model providers. Here, coordination is useful — HCP can get memories only if providers expose export mechanisms — yet this coordination need not be voluntary. For example, if one model provider offers such an export mechanism, competitive forces may induce others to follow. In this way, public interest in building HCP can itself generate the coordination needed to enable it.
>
> Second, adoption has network effects. As more services integrate with HCP, the preference layer becomes more valuable. So, even if HCP is rolled out for a particular niche (e.g., agents that store and scope one's travel preferences), once users have it, the marginal cost of extending it to adjacent domains is low for both users and integrating agents. Adoption can grow domain by domain, with each partial integration making the whole more valuable (§4.4.1).
>
> Third, on regulation: we are not advocating for a specific regulatory intervention, but existing frameworks may already point in the direction of HCP. GDPR Article 20 gives users the right to data portability, but for AI preference data this right is largely unexercised today, as there is no standard format or endpoint to receive the data into. In that light, HCP is infrastructure that makes that right exercisable, and existing regulations may aid HCP in obtaining early buy-in from incumbents. Further, the success of open banking from PSD2 is instructive not as a call for mandates, but as evidence that when the infrastructure exists, regulation can resolve incumbent resistance effectively (Babina et al., 2025). We see HCP as building that infrastructure first.
>
> **W2: Why not present requests directly to the user instead of delegating to the HCP model?**
>
> User friction has been a primary obstacle to widespread adoption of personal data stores. Systems that require explicit approval for every request place too high a burden on the median user in practice. As such, HCP delegates routine decisions to the HCP LLM precisely to reduce this friction. That said, users who want more direct control can configure HCP to require explicit per-request authorizations for their preference data (§3.2.4). The design is intended to make this tradeoff the user's to set.

---

> > ### Author Rebuttal · Reviewer_9ZGe · 2026-04-03
> >
> > Thanks for answering the questions I raised. These were not a blocker for in recommending accepting the paper, and since my score is already positive I will keep my score as is.

---

### Official Review · Reviewer_CtCN · 2026-03-12

**Significance:** 2
**Argument Clarity:** 2
**Rating:** 4
**Confidence:** 3

**Questions:**

1. The paper acknowledges that more capable external models might manipulate the HCP LLM into oversharing. What concrete defenses do you propose? Have you considered formal verification of the minimization policy?

2. I think your password manager analogy breaks down because HCP requires provider-side integration. Given that the open banking analogy you cite required a regulatory mandate (PSD2), do you view regulation as a necessary condition for HCP adoption?

3. Users may express contradictory preferences over time or across domains. How should the HCP LLM resolve conflicts? What happens when conflict resolution produces outcomes the user did not intend?

4. When a guardian's policies conflict with a user's expressed preferences (e.g., an employer guardian restricting an employee's data sharing), what adjudication mechanism/policy do you envision?

**Alternative Views Section:**

Yes

**Compliance With Llm Reviewing Policy A Conservative:**

Affirmed.

**Discussion Potential:**

2

**Final Justification:**

I am satisfied with their response. I increased the score

**Paper Summary:**

This is a position paper that argues that current AI personalization, where user preferences are inferred from behavioral data and grouped within individual providers, is inadequate for a future in which LLMs act as general-purpose assistants. The authors propose the Human Context Protocol (HCP), a user-governed intermediary layer that stores preferences in natural language, enforces scoped and revocable access permissions, and enables portability across AI services. The paper situates HCP within the lineage of personal data sovereignty initiatives (infomediaries, MyData, Solid, SSI, Web3 wallets), contrasting them in Table 1. A system design is outlined covering context representation, CRUD actions, initialization, and access control (OAuth 2.0, encryption, audit logs), accompanied by an open-source proof-of-concept. Section 4 engages with four counterarguments: the Coasian objection (market efficiency suffices), the revealed preference objection (inference is enough), the model-level objection (fine-tuning/steering suffices), and practical challenges around standards convergence and deep personalization risks. A call to action for ML researchers and policymakers is presented, with an appendix discussing applications in scheduling, health, education, matching markets, democratic governance, and guardian assistant systems.

**Position:**

Yes

**Position In Title:**

Yes

**Related Work:**

2

**Strengths And Weaknesses:**

Strengths:
1. The paper's use of market failure analysis to argue why the Coasian benchmark fails for preference data is the strongest intellectual contribution. The four sub-arguments (lock-in/switching costs, non-rivalry of data, information asymmetries, and failure of market thickness for heterogeneous preferences) are well-grounded in economic theory and individually cited.

2. The alternative views section is substantive. The revealed preference objection is good, connecting to Kleinberg et al.'s "inversion problem" and behavioral economics literature on mental accounting and self-control failures. The model-level objection correctly identifies the portability limitation of per-provider fine-tuning.

3. The open-source prototype is simple, and demonstrates feasibility and lowers the barrier for follow-up work. The end-to-end Alex scenario is concrete and effective at communicating the intended workflow.

Weaknesses:
1. The system design remains at a high level of abstraction. Key design tensions are not fully resolved or even explored in detail. How does the HCP LLM perform data minimization reliably? What happens when the HCP LLM itself is manipulated by adversarial queries from more capable external models?

2. While the paper correctly identifies that incumbents have weak incentives to adopt HCP, the proposed solution — a password-manager-style organic adoption path — is not convincingly argued. Password managers work via passive browser integration, requiring no cooperation from services. HCP, by contrast, requires active API/MCP integration from providers. The paper acknowledges this difference but then argues that starting with MCP-compatible assistants will bootstrap adoption. This is the central challenge for HCP's viability, and it deserves much more rigorous analysis.

3. The paper lists security requirements (encryption at rest/in transit, OAuth 2.0, audit logs, zero-trust architecture) but these are standard practices, not novel contributions. The genuinely novel security challenges — adversarial prompting attacks against the HCP LLM, inference attacks that reconstruct preferences from outputs, and the integrity of a smaller HCP LLM mediating between a user and more powerful external models — are mentioned as "emerging threats" but receive no substantive treatment.

4. I think, even for a position paper, some form of evaluation would strengthen the argument. A user study on willingness to adopt preference management tools, a simulation of preference portability benefits, or even a systematic comparison of preference representation formats would greatly enhance the position paper.

**Support:**

2

---

> ### Author Rebuttal · Authors · 2026-03-31
>
> ## Author Response to Reviewer CtCN
>
> Thank you for the review. We have directly answered your questions below.
>
> **Q2: Adoption**
>
> We read your concern as: past user-controlled data infrastructure has struggled, the closest success (open banking) required regulation, and HCP needs provider cooperation. Why expect adoption here?
>
> We have three points.
>
> 1) HCP may not require provider cooperation in the way the review suggests. HCP sits after the base model, between it and downstream agents, connected along MCP. A downstream agent requesting shopping preferences does not care whether they come from the model's memory or from an HCP — it just requests the preferences. The adoption question is therefore less about convincing providers, and more about whether user-mediated personalization provides value that the status quo does not.
>
> 2) On value: prior efforts on personal data control may have struggled due to high user friction and low expected payoff from personalization. LLMs change both terms of this cost-benefit analysis — the value from personalization is substantially larger when the downstream technology is a general-purpose assistant, and preferences that are hard to set as structured fields can be easier to say in natural language (e.g., "I prefer window seats but not on red-eyes").
>
> 3) On regulation: PSD2 improved competition and user outcomes (Babina et al., 2025), and regulation may eventually prove useful here too — though we think this is largely empirical, and premature mandates on a fast-moving frontier risk doing more harm than good. Critically, PSD2 succeeded because the infrastructure already existed for regulators to mandate. We are in earlier days: while one day much economic activity may be mediated by LLM agents, today we do not yet have a paradigm for personalization of the personal data that empowers those transactions. Our paper aims to foster that conversation.
>
> **Q1: Adversarial prompting and formal verification**
>
> We agree that adversarial prompting against the HCP LLM is a genuine and important concern, and that formal verification is a natural method to consider here. However, we also believe that addressing this problem requires longer-term community effort — it is an instance of a broader open problem of formally verifying LLM behavior, which remains unsolved. Our goal in this paper is not to solve all the challenges that HCP raises, but to define a set of problems clearly enough to inspire new research. The security and robustness of LLM-mediated access control is one such direction, and recent work has begun to engage with related privacy risks in LLM agent settings (e.g., [https://arxiv.org/pdf/2508.10880](https://arxiv.org/pdf/2508.10880)).
>
> **Q3 + Q4: Preference conflicts and guardian adjudication**
>
> On Q3: it may be that apparent inconsistencies are not conflicts at all, but simply preferences that apply in different contexts (e.g., I like loud concerts but quiet libraries). However, when a genuine within-domain conflict arises (e.g., a user expressed a preference for budget airlines via ChatGPT last year but recently indicated they value comfort via Gemini), versioning can resolve by recency or confidence scores, and the HCP can surface when an older preference appears stale or in tension with recent statements, prompting the user to confirm or update. How well this works likely depends on both the HCP's design and the intelligence of the underlying model. We see this ability to surface inconsistencies across interactions for user review as a distinct benefit of interoperable preference infrastructure.
>
> On Q4: where the default for within-domain conflict resolution is recency, for guardian conflicts it is hierarchy. The user's preferences hold by default, but guardian policies — set by employers, parents, or institutions — would override them where applicable, similar to a "SafeSearch" filter on Google. This hierarchy can be as complex as an organization requires: different roles may have separable authority over different categories of preference data, and some categories may require approval from multiple parties before being shared externally.
>
> We discuss guardian systems in the Appendix and will sharpen the adjudication mechanism along these lines for camera-ready.

---

> > ### Author Rebuttal · Reviewer_CtCN · 2026-04-01
> >
> > I am satisfied with their response to most of my questions. However, discussions on how to tackle the adversarial prompting and formal verification are unsatisfactory. For any HCP, this is quite important. I will adjust my score accordingly for the final justification.

---

### Official Review · Reviewer_C4pj · 2026-03-13

**Significance:** 3
**Argument Clarity:** 3
**Rating:** 5
**Confidence:** 4

**Questions:**

1. The paper discusses preference “Initialization” (Section 3.2.3), noting that current personalization relies on behavioral proxies such as “clicks, time spent, purchase history” (L367-370). In contrast, the authors suggest that bootstrapping could use “imported digital traces such as playlists and browsing history” (L230-231). How is this approach meaningfully different from the existing behavioral proxies?
2. If HCP is implemented on top of an AI provider that already maintains its own memory layer, how would potential conflicts be resolved? For example, a user might want to stop sharing certain preferences via HCP, but the provider’s internal memory could still retain that information. How would consistency and user control be ensured in such cases?
3. HCP emphasizes explicit user authorization for sharing personalizations. How does the protocol handle the tradeoff between automatically inferred preferences and user control? Could inferred preferences still be incorporated alongside HCP, particularly for users who may not want to specify what is shared item by item? What is the minimum information users would need to provide for HCP to function effectively?
4. There is a potential tradeoff between helpfulness and personalization control. Current AI providers often gradually infer preferences to improve helpfulness. To what extent might HCP limit this functionality, and how should the system balance user control with the expectation that AI systems gradually learn users’ preferences over time?

**Alternative Views Section:**

Yes

**Compliance With Llm Reviewing Policy A Conservative:**

Affirmed.

**Discussion Potential:**

3

**Final Justification:**

My concerns are fully resolved. Since my score is already positive, it remains unchanged.

**Paper Summary:**

This paper argues for the need for a Human Context Protocol (HCP), proposed as the human counterpart to the Model Context Protocol (MCP). HCP is presented as a key requirement for enabling robust AI personalization. Similar to MCP, HCP aims to provide a standardized protocol through which AI assistants can interface with user-specific personalization data.

The core ideas behind HCP include:
(1) portability of user personalization across different AI models, (2) the ability to capture user preferences in rich and flexible formats (e.g., text, graphs, vectors, etc.), (3) user control over what information is shared, and (4) strong authorization mechanisms, ensuring that AI systems can access only the subsets of preferences explicitly permitted by the user.

Although the paper does not prescribe a definitive implementation, it outlines a potential protocol architecture. This includes: (1) mechanisms for user preference representation, (2) RESTful-style operations similar to those used in MCP, (3) methods for initializing user preferences for new users (e.g., bootstrapping, leveraging existing documents, etc.), and (4) authorization and security mechanisms.

In addition, the paper: (1) presents an anonymized open-source prototype demonstrating a proof of concept, (2) provides a concrete example workflow illustrating how HCP could function in practice, (3) engages with alternative perspectives on the proposed approach, and (4) concludes with a broader call to action for the community to explore this direction.

**Position:**

Yes

**Position In Title:**

Yes

**Related Work:**

4

**Strengths And Weaknesses:**

Strengths
1. The paper addresses a timely and important concern regarding user privacy, ownership, and control in how AI model providers implement personalization.
2. The author clearly articulates the limitations of current provider-centric personalization, where each provider maintains a partial “memory” of the user that lacks transparency, portability, and user control, highlighting an issue that is increasingly relevant as personalization becomes more integrated into AI systems.
3. The paper provides an extensive scenario example illustrating how HCP could function in practice, along with a workflow describing how AI systems might interact with user preference data.
4. The authors present an anonymized open-source prototype demonstrating a proof of concept, helping readers move beyond the abstract idea of a protocol and see how it could be implemented in real-world systems.
5. The paper discusses alternative approaches (e.g., inferred preferences, fine-tuning/steering), which help contextualize HCP and clarify the motivations behind the proposed protocol.
6. The authors acknowledge limitations, risks, and practical challenges, including considerations around implementation, adoption, and feasibility in real-world ecosystems.

Weaknesses
1. The paper could provide more discussion on preference initialization, as determining a user’s initial preferences may be a nuanced and challenging task.
2. It would be helpful to clarify how HCP might operate when layered on top of AI providers that already maintain their own memory layers, as this raises questions about potential inconsistencies in memory sharing (e.g., user stops sharing a subset of their preferences, but the AI provider retains them).
3. The tradeoff between explicit user control and inferred preferences could be explored further, particularly regarding whether HCP should allow automatically inferred preferences or rely entirely on information explicitly authorized and stated by the user.
4. A more detailed discussion of the potential impact of HCP on the helpfulness of AI systems could also strengthen the paper, especially concerning how limiting access to inferred or gradually learned preferences might influence system performance.

**Support:**

3

---

> ### Author Rebuttal · Authors · 2026-03-31
>
> Thank you for the comments. We are glad the alternative views section, the prototype, and the practical limitations discussion resonated. We've offered direct responses to each of your questions below.
>
> **Q1: How is bootstrapping from "imported digital traces" different from existing behavioral proxies?**
>
> It is true that HCP would like to use imported digital traces from chats with model providers or other digital traces like browsing history. The difference between the two approaches is primarily in governance and control of the underlying data. Users often retain formal ownership of their inputs, but in consumer AI products, providers maintain practical control over user data. Providers control what preferences get inferred from traces and how those inferences are used, and users have little recourse to see, correct, or move the resulting preferences elsewhere. Under HCP, the same traces are converted into natural-language preference statements that the user controls: readable, editable, and portable across providers.
>
> Further, some providers do already offer memory features with partial control over stored preferences (e.g., ChatGPT allows for viewing and deleting memories, but not editing), but these remain provider-specific and non-portable – a user's preference state in Gemini cannot inform their experience in Claude, or vice versa.
>
> **Q2: How would HCP handle conflicts with a provider's own memory layer?**
>
> Thank you for the question, and we will add discussion of this in §4.4 for camera-ready.
>
> The natural approach is for HCP to serve as the last layer between users and downstream agents. Even if a provider's model produces its own personalization, that output would pass through HCP, which enriches or scopes it before reaching the downstream agent.
>
> That said, it is fair to note that these two systems (the HCP LLM and the base model) may differ in how they represent and act on preferences. We think this is actually an argument for HCP rather than against it. Today, when a provider's internal memory diverges from what a user really wants, there is limited mechanism to surface that disagreement. If a user has indicated that preference in the past via another model or directly to the HCP, the HCP makes these divergences visible and can then learn from them.
>
> Moreover, this conflict handling is also why HCP's value to users grows with every additional service that integrates with it. This creates natural adoption incentives, particularly for new entrants, and can be further facilitated by lowering integration costs with features like single sign-on.
>
> **Q3: How does HCP handle the tradeoff between inferred preferences and explicit user control? What's the minimum info needed?**
>
> We see HCP as complementing inferred preferences, not replacing them (§4.2). As users use a particular model, providers can continue inferring preferences from behavior. And across all siloed models a user may use, HCP is one way to aggregate, correct, or share those inferences across models. If a user wants more control, they can simply control how widely their preference data is scoped. E.g., "don't share any information in the health category unless with a medical provider."
>
> On minimum information: the minimum viable HCP is an empty profile that accumulates context over time, though in practice a "warm start" from importing existing preference data from a base model's memories may be easier (§3.2.3 and e.g., [new export rights with Claude](https://support.claude.com/en/articles/12123587-import-and-export-your-memory-from-claude)). A simple intake process may also be a low cost way to populate the HCP with the most important preference data should the cold-start problem be serious in practice. Over time, the profile should fill in organically as it aggregates preference information from various model interactions.
>
> **Q4: To what extent might HCP limit helpfulness by restricting gradual preference learning?**
>
> We actually think this is a feature rather than a problem. Today's systems optimize helpfulness with no user-defined guardrails, and that is part of what produces the sycophancy and manipulation risks we discuss in §4.4.2. HCP lets users define their own tradeoff — someone who wants maximum helpfulness can configure HCP to default to sharing everything, while someone who wants tighter control can scope permissions more narrowly. The point is that this tradeoff should be the user's to make, not the provider's. And importantly, it is reversible, as users can loosen or tighten permissions at any time so there is no irreversible cost to starting with more control.

---

> > ### Author Rebuttal · Reviewer_C4pj · 2026-04-03
> >
> > Thank you for engaging with my questions thoughtfully, especially Q4. I encourage the authors to incorporate the rebuttal in the revision to strengthen the paper. Since my original score is already positive, I will keep it.

---

### Official Review · Reviewer_oiSX · 2026-03-15

**Significance:** 3
**Argument Clarity:** 2
**Rating:** 3
**Confidence:** 3

**Questions:**

- Could the authors clarify how locally stored user data would be structured, shared, and accessed across different services and AI agents?
- What mechanisms would ensure both interoperability and privacy in this proposed framework?

**Alternative Views Section:**

Yes

**Compliance With Llm Reviewing Policy A Conservative:**

Affirmed.

**Discussion Potential:**

3

**Paper Summary:**

This position paper argues that the rise of large language models creates an opportunity to redesign personalization around greater user control, interpretability, and portability. Today’s personalization systems are typically managed by service providers, which infer preferences from users’ behavioral data and keep those profiles locked within platform-specific infrastructures. In contrast, the authors propose a new **Human Context Protocol (HCP)** that represents personal preferences as a portable, user-governed layer that can be shared across AI systems. Rather than relying mainly on opaque behavioral inference, HCP emphasizes preferences expressed through natural language and contextual information, enabling users to manage how their preferences are represented, accessed, and revoked. The paper argues that this approach could support more trustworthy and user-centric personalization in the emerging human–AI economy. To ground the discussion, the authors present a working prototype and examine practical challenges, including adoption incentives, market dynamics, and risks in high-stakes settings. Overall, the paper presents HCP as a foundation for interoperable, controllable personalization.

**Position:**

Yes

**Position In Title:**

Yes

**Related Work:**

3

**Strengths And Weaknesses:**

Strengths:
- I like the idea of building a user-centered preference layer. It has the potential to better protect user data while also improving preference alignment across systems.

Weaknesses:
- The paper proposes storing user data on local devices, but it remains unclear how such data would be stored, managed, and used across different services and AI agents.
- In particular, the paper could explain more concretely how interoperability would work in practice while still preserving privacy, usability, and user control.

**Support:**

2

---

> ### Author Rebuttal · Authors · 2026-03-31
>
> Thank you for the comments. We are glad you support the core idea. Below we answer your questions directly.
>
> **Q1: How would locally stored user data be structured, shared, and accessed across different services and AI agents?**
>
> §3.2 addresses this in more detail, but we highlight the key points here.
>
> On storage: there are many potential ways to manage user data across services. One of the simplest ways is to save everything in markdown (e.g., memory.md) and use an LLM to manage memory updates and context retrieval – though the actual implementations could vary dramatically, from JSON or markdown files, to a key-value store, to integration with existing personal data pods like Solid (§3.2.1).
>
> On structure: preferences are stored as natural-language statements enriched with lightweight schema annotations (category tags, confidence scores, versioning) to reduce ambiguity and track how preferences update over time.
>
> On sharing/access: HCP exposes a standard set of operations (read, write, update, delete) as RESTful endpoints or MCP tool calls, making it accessible to any MCP-compatible agent without bespoke integration (§3.2.2). Scoped access control means each agent receives only the preference categories it is authorized to see.
>
> Finally, we point towards one implementation via our open prototype (footnote 1). We emphasize that there are many possible implementations of HCP, and that the best ones are yet to be built.
>
> **Q2: What mechanisms would ensure both interoperability and privacy?**
>
> These two desiderata are addressed by different parts of the design. Interoperability comes primarily from natural language, as preferences stored as text are natively readable by any LLM agent or base model. Privacy (beyond that from encryption and OAuth, §3.2.4) comes from scoping preference data: the HCP intermediary enforces minimal disclosure at inference time, so, for example, a recipe agent sees culinary preferences but not health data when helping one bake a cake.

---

> > ### Author Rebuttal · Reviewer_oiSX · 2026-04-01
> >
> > Thank you for the clarification. The rebuttal better explains possible storage formats, lightweight schema annotations, and scoped access through REST/MCP interfaces. However, key implementation details remain underspecified, especially around cross-service synchronization, conflict resolution, and concrete privacy guarantees beyond scoped disclosure. My concerns are reduced, but not fully resolved.

---

### Decision · Program_Chairs · 2026-04-30

**Decision:**

Accept (regular)

**Comment:**

I recommend acceptance. The paper presents a timely, distinctive, and discussion-worthy position at the intersection of personalization, user control, and AI systems, and the review set is overall positive after rebuttal, with several reviewers explicitly finding their concerns fully or substantially resolved and highlighting the clarity of the position, the strength of the alternative-views section, and the value of the prototype in making the proposal concrete.